# Trends in Homicide Hospitalization and Mortality in Taiwan, 1998–2015

**DOI:** 10.3390/ijerph19074341

**Published:** 2022-04-05

**Authors:** Shih-Chun Hsing, Chu-Chieh Chen, Shi-Hao Huang, Yao-Ching Huang, Bing-Long Wang, Chi-Hsiang Chung, Chien-An Sun, Wu-Chien Chien, Gwo-Jang Wu

**Affiliations:** 1Center for Healthcare Quality Management, Cheng Hsin General Hospital, Taipei 11220, Taiwan; ch8363@chgh.org.tw; 2Department of Health Care Management, College of Health Technology, National Taipei University of Nursing and Health Sciences, Taipei 11220, Taiwan; chuje@ntunhs.edu.tw; 3Graduate Institute of Life Sciences, National Defense Medical Center, Taipei 11490, Taiwan; 4Department of Chemical Engineering and Biotechnology, National Taipei University of Technology (Taipei Tech), Taipei 10608, Taiwan; hklu2361@gmail.com (S.-H.H.); ph870059@mail.ndmctsgh.edu.tw (Y.-C.H.); 5Department of Medical Research, Tri-Service General Hospital, National Defense Medical Center, Taipei 11490, Taiwan; g694810042@gmail.com; 6School of Public Health, National Defense Medical Center, Taipei 11490, Taiwan; billwang1203@gmail.com; 7Taiwanese Injury Prevention and Safety Promotion Association, Taipei 11490, Taiwan; 8Big Data Research Center, College of Medicine, Fu-Jen Catholic University, New Taipei City 242062, Taiwan; 9Department of Public Health, College of Medicine, Fu-Jen Catholic University, New Taipei City 242062, Taiwan; 10Obstetrics and Gynecology Department, Tri-Service General Hospital, Taipei 11490, Taiwan; 11Graduate Institute of Medical Sciences, National Defense Medical Center, Taipei 11490, Taiwan

**Keywords:** homicide, epidemiology, hospitalized patient, national health insurance data

## Abstract

In Taiwan, the national research on homicide is rare, mostly discussing the issue of child abuse. We sought to better understand the characteristics and risk factors of homicide through a retrospective cohort study from 1998 to 2015. “Child battering and other maltreatment” ranked first for the 0–4 age group and second for the 5–14 age group. The hospital mortality was 511 deaths. We found that the 25–44 age group had the highest risk and accounted for 44.76% of hospitalization. The most common causes were “fight, brawl, and rape” (49.12%), “cutting and piercing instruments,” (13.16%) and “child battering and other forms of maltreatment” (4.72%). Additionally, the percentages of “fight, brawl, and rape,” “firearms and explosives,” and “cutting and piercing instruments” were significantly higher among males than among females. The percentages of “hanging and strangulation,” “corrosive or caustic substance,” “child battering and other maltreatment,” “submersion,” and “poisoning” were significantly higher among females than males. Factors associated with homicide in-hospital mortality included gender, age, low income, catastrophic disease, Charlson comorbidity index score, urbanization level, hospital level, classification of hospitalization, and surgery. Overall, the trend of hospitalization rates due to homicide decreased both by gender and age group, except for the 0–4 age group: cause of homicide first, hanging and strangulation second, firearms and explosives third; type of injury, hospitalized patients with “vascular injuries” first, joint and muscle sprain, and intracranial, chest, and abdominal pelvic injuries second, and “burns” third with a higher risk of death. Homicide reduction requires a comprehensive strategy beyond specific victim groups. Interagency collaboration should be strengthened, especially between law enforcement/criminal justice and public health.

## 1. Introduction

Violence is a significant public health issue. The World Health Organization (WHO) reported that approximately 470,000 homicides occur annually, and millions of people suffer violence-related injuries. Beyond death and injury, exposure to violence can increase the risk of health problems [1]. WHO defines homicide as the killing of a person by another with intent to cause death or serious injury, by any means. The forms include child maltreatment, youth violence, intimate partner violence, sexual violence, elder abuse, and so on. Data from WHO showed that the 2015 homicide rate was 6.4 per 100,000 population, 80% of homicides occurred in males, and the highest rates were in males aged 15–29 years [2]. Recently, homicide prevention has been considered an important task for the sustainable development goal internationally [3].

In 2019, the United States had 19,141 homicide deaths, and many more were victims of non-fatal violent injuries or violent crimes [4,5]. Exposure to violence—including direct victimization, witnessing, and hearing violence, as well as living in areas where violence is common—is associated with enormous health and social costs (e.g., premature death, depression, and reduced economic productivity) [6]. Exposure to property crime (e.g., burglary, and theft) may also decrease health and well-being by affecting feelings of security, subsequently affecting mental state, outdoor physical activity, and social capital negatively [7]. A theoretical explanation for the consistently high southern homicide rates is that the South constitutes a “regional culture of violence”. One study which analyzed data from the 67 counties in Florida supports the notion that the differential distribution of medical resources is partially responsible for the variation in criminally induced lethality rates [8,9]. Advances in emergency medical care have significantly and increasingly reduced the lethality of violent assaults [10,11,12].

Once a homicide case occurs in Taiwan, the breaking news is reported repeatedly, causing panic but no relevant measures to soothe people. Besides, the national research on homicide is rare, mostly discussing the issue of child abuse. However, homicide is not the only involved in child abuse. Homicide can occur in all ages, such as domestic violence and gang war, resulting in appalling social events that shock the entire country.

Yu’s study examined the homicide mortality trend and showed that a total of 7419 people (male: 5488, 73.97%; female: 1931, 26.03%) died of homicide between 1986 and 2007 in Taiwan [13]. The mortality rate was 1.56/10^5^ (male: 2.27, female: 0.84). The common homicide types were assault by “cutting and piercing instrument” (51.45%), “firearms and explosives” (11.46%), and “hanging and strangulation” (6.67%). Death by “hanging and strangulation” was the only type of homicide that had higher numbers and a higher mortality rate among females (0.14/10^5^) than males (0.08/10^5^); as well as the highest mortality rate for younger children (0–4 age group) [13]. The homicide mortality rate had a declining trend, except for the 0–4 age group [13].

Lai’s research showed that the total injury mortality rate declined, but motor vehicle injury, drowning, and fall injury were three unintentional injuries of concern. Additionally, suicide and child homicides are key points to prevent in the future [14]. Chien’s study showed that after 1989, the mortality rates for unintentional injuries and suicide declined, but the homicide rate for children increased [15]. Therefore, laws to prevent violence in homes must be enforced, and drowning prevention programs should be implemented and incorporated into the Children and Adolescent Safety Implementation Program. Preventive efforts should also target MVI and suicide in the 15–19 age group, drowning at all ages, and suffocation and homicide in infants and children five years old [15,16]. Therefore, this study utilized the data from the Health and Welfare Data Science Center of the Ministry of Health and Welfare to collect moderate injury-level patients (inpatients) as study cases to understand the epidemiological characteristics of hospitalization due to homicide and inpatient mortality-related factors in Taiwan for 1998–2015. We discussed trends of gender and age in homicide (hospitalization) and compared causes of hospitalization and injury types due to homicide and assault, and relevant factors of inpatient mortality due to homicide.

## 2. Materials and Methods

### 2.1. Data Sources

The National Health Insurance program of Taiwan covers 99% of Taiwan’s population (23 million people). Thus, the data from HWDC are representative of the entire population for empirical health and medical research. This study used the 1998–2015 data on inpatient expenditures by admission (DD files) and registry of contracted medical facilities (HOSB files) for analysis (TSGHIRB number 1-105-05-142).

### 2.2. Variable Definitions

Variables included gender (male and female), age (0–4, 5–14, 15–24, 25–44, 45–64, and ≥65 years), low income (yes, or no), catastrophic illness (with or without), Charlson comorbidity index (CCI), season (spring, summer, autumn, and winter), hospitalization area (northern, central, southern, eastern, and outer islands), urbanization level (high, medium, and low), hospital level (medical center, regional hospital, and local hospital), hospitalized classification (general medical, general surgery, psychiatry, and others), surgery (with or without), hospitalization days (day), homicide causes (ICD-9-CM E-Code and E960-E969), injury types (ICD-9 Codes 800–999, 905–909, and 958–958 excluded), medical expense (USD), and prognosis (survival or mortality).

The CCI selects the first five diagnostic codes (ICD-9-CM N-Code), weighs them according to scoring criteria defined by Charlson, and calculates the total score [17]. Higher scores indicate more complications or a more severe diagnosis. Additionally, the prognoses for the deceased include death in the hospital and voluntary discharge for the terminally ill.

### 2.3. Statistical Analysis

Data analyses were performed using SPSS software for Windows, version 22.0 (IBM Corp., Armonk, NY, USA). The analytic content was described as follows: Statistical methods, including frequency distribution, percentage, mean value, and standard deviation were used to describe all epidemiological characteristics of homicide hospitalization in Taiwan. Additionally, logistic regression was used to discover factors related to inpatient mortality from homicide. A *p*-value < 0.05 was set as the standard of statistical significance.

## 3. Results

The 95,878 hospitalized victims of homicide included 74,991 (78.22%) males and 20,887 (21.78%) females. The mean annual number of hospitalizations was 5327 people. The inpatient mortality totaled 511 people (male: 82.58%, female: 17.42%), an average of 28 people annually. The inpatient mortality rate was 0.53%. The total inpatient rate was 23.48/10^5^, with more males than females (36.18/10^5^ vs. 10.40/10^5^).

The overall hospitalization trend showed that 2000 was the peak period (the same in both males and females), and then gradually decreased. Males had a higher hospitalization rate than females (Figure 1).

The age group of 15–24 years had the highest hospitalization rate (per 100,000) among all age groups, followed by the 25–44-year group. However, except for the children aged 0–4 years, the hospitalization rates in the other age groups all showed a decreasing trend (Figure 2).

The mean age of the patients hospitalized due to homicide and assault was 36.38 years (40.34 years among females, significantly older than 35.28 years among males). The 25–44 age group had the highest percentage of hospitalizations, both male and female (male: 44.76%; female: 47.68%). The percentages of the 5–14 and 15–24-year age groups were higher among males than among females. Conversely, the percentages of the 0–4-year group and older than 25-year groups were significantly higher among females than males. Most of the patients hospitalized for homicide belonged to non-low-income households. The proportion of hospitalizations was highest in autumn for men and in summer for women. The proportion of inpatients in the central region was the highest for both males and females. The urbanization level of hospitalization for both men and women were the highest in the medium ratio. The ratio of inpatient hospitals for both men and women were the highest among regional hospitals.

General surgery has the highest proportion of inpatient departments for both males and females. Most hospitalizations for men and women do not need surgery. The average number of hospitalizations for men and women was 1.10 ± 0.43. The average length of hospital stay for men and women was 5.86 ± 10.60 days. The average medical cost for men and women was 983.9 ± 2286.4 (USD). Most men and women had a good prognosis after hospitalization (Table 1).

### 3.1. Comparison of Causes of Hospitalization Due to Homicide and Assault

The percentage of homicide caused by “firearms and explosives” (0.82%/0.28%), “cutting and piercing instruments” (14.49%/8.40%), and “fight, brawl, and rape (49.45%/47.94%) were 2.93, 1.73, 1.03 times higher among males than females, respectively. Men were at greater risk than women for “firearms and explosives.” The percentages of other causes among females were significantly higher than those among males with “hanging and strangulation” (0.13%/0.03%), “corrosive or caustic substances” (0.27%/0.07%), “child battering and other maltreatment” (10.86%/3.02%), “poisoning” (1.27%/0.49%), and “submersion (drowning)” (0.05%/0.02%) 4.33, 3.86, 3.60, 2.59, and 2.50 times higher, respectively. Women were at greater risk than men were for “hanging and strangulation” (Table 2).

### 3.2. Comparison of Injury Types of Homicide Hospitalization

Among the 95,878 patients, the percentages of “fractures of the skull, trunk, and upper and lower limbs” (34.25%/21.76%), “vascular injuries” (1.64%/0.88%), “open wounds” (51.92%/29.65%), “crush injuries” (0.45%/0.30%), “spinal cord injuries” (3.6%/2.42%), and “burns”: (0.05%/0.04%) were 15.73, 1.86, 1.75, 1.5, 1.48, and 1.25 times higher among males than females, respectively. Men were at greater risk than women for “fractures of the skull, trunk, and upper and lower limbs.” Conversely, the percentages of “drug and substance poisoning” (1.97%/0.52%), “superficial skin contusions” (49.17%/39.0%), “sprains of joints and muscles” (2.18%/1.85%), and “intracranial, thoracic, and abdominal pelvic injuries” (56.31%/48.30%) were 3.78, 1.26, 1.17, and 1.16 times higher among females than males, respectively. Women were at greater risk than men for “drug and substance poisoning” (Table 2).

### 3.3. Relevant Factors of Inpatient Mortality Due to Homicide

After other factors were controlled, the relevant factors of inpatient mortality due to homicide included gender (1.765 times higher risk in males than females), age (1.6% increase in death risk yearly), low income (3.014 times for non-low income), catastrophic illness (6.741 times for without), CCI score (18.4% increase in death risk for per point increase in score), urbanization level (1.666 times higher risk in high than in low), hospital level (3.201 times higher risk in medical center than district hospital), hospitalized classification (3.986 times higher risk in general medical than others), surgery (1.398 times higher risk in with than without), homicide cause (submersion (drowning) (84.901 times), hanging and strangulation (42.454 times), and firearms and explosives (5.332 times)), and injury types (the inpatients with “vascular injuries,” (4.397 times for without), sprains of joints and muscles, intracranial, thoracic, and abdominal pelvic injuries (3.151 times), as well as “burns” (2.896 times) had a higher mortality risk) (Table 3).

## 4. Discussion

### 4.1. Trends of Gender and Age in Homicide (Hospitalization)

Our study showed that of the 95,878 people hospitalized due to homicide between 1998 and 2015 in Taiwan (male: 78.22%, female: 17.42%), 511 people had inpatient mortality (male: 82.58%), corresponding to an inpatient mortality rate of 0.53% (the mortality among males was 1.3 times higher than females (36.18/10^5^ vs. 10.40/10^5^)). The homicide rate in Taiwan was similar to that in other countries and primarily occurred among males. Additionally, the results in this study also showed that injuries among males were significantly more severe than in females. Therefore, the percentages of subsequent inpatient mortality, mean hospitalization days, and mean medical expenses of males were all higher than those of females. An estimated 464,000 people were victims of intentional homicide in 2017 [18]. The global average homicide rate in 2017 was estimated at 6.1 victims per 100,000 people, and about 90% of all recorded homicides worldwide are committed by male perpetrators [18,19]. Men account for nearly 80% of all recorded homicide victims worldwide [18,19,20]. Literature from other countries noted that homicide victims are mainly males. Studies in the United Kingdom, Malaysia, and China showed that 66%, 89%, and 76% of homicide victims were males, respectively, which is consistent with our study [18].

This study showed that the percentage of hospitalization due to homicide was highest among the young adult population aged 15–24 years, followed by the adult population aged 25–44 years. Thus, homicide hospitalization in Taiwan mainly occurred among young adults and adults. These two populations account for 68.9% of homicide hospitalizations. It also confirmed that violence (homicide) was a universal issue among the young adult and adult populations. Furthermore, a study analyzed the long-term trend of the homicide mortality rate between 1986 and 2007 in Taiwan: the results found that the homicide mortality rates among all age groups showed a decreasing trend. However, mortality among children aged 0–4 years (particularly among female children) showed an increasing annual trend [13], similar to our study results. The above results showed that the mortality and hospitalization rates of homicide among children aged 0–4 years did not decrease in the last three decades in Taiwan. Therefore, government departments and social farewell units must take strong actions targeting the prevention and control of homicide among children and propose complete intervention plans to impede the continuous aggravation of homicide among children.

### 4.2. Homicide (Hospitalization) Causes

This study indicated that “fight, brawl, and rape” were the top causes of hospitalization due to homicide in the past 18 years in Taiwan and accounted for 49.12% of all causes. This rate was significantly higher than others (such as “cutting and piercing instruments,” accounting for 13.16%, and “child battering and other maltreatment,” accounting for 4.72%) and might be associated with the accessibility of tools. Additionally, a study in the UK noted that the kitchen knife was the most frequently used tool in homicide cases mainly because it was very common and accessible in daily life [21]. A recent UK study independently investigated and found that the rate of use of kitchen knives in homicides by mentally ill offenders is high, presumably being the sharpest tool, most commonly used, and being significantly less common among homicide-related unplanned households attacks targeting known victims [22]. This result also explains why injury caused using “cutting and piercing instruments” (items accessible in daily life, such as knives and sharp objects) was the second highest cause of hospitalization due to homicide in Taiwan.

This study showed that “child battering and other maltreatment” ranked the third highest cause of hospitalization due to homicide (4.72%). There were 4530 abused inpatients. After 3022 people who non-specific people abused were excluded, the top three causes of the other 1549 abused inpatients were abuse by spouses (49.26%), abuse by fathers (26.28%), and abuse by other relatives (6.71%). The above data show that females accounted for 95.81% of patients abused by spouses, similar to the analysis in other countries showing that victims were mainly females and were mostly killed by spouses with close relationships [23]. Domestic violence can include violence between husbands and wives, girlfriends and boyfriends, or gay partners. It could be violence between parents and children, between adult children and aging parents, or between siblings [24]. The other study noted that 90% of homicide cases among children aged 0–4 years were committed by one of their parents or other family members with a close relationship [25], consistent with this study (people who were abused by caregivers of non-relatives only accounted for 3.42% of abused subjects). These results were also consistent with a Swedish study describing 90 children aged 0–14 years who were killed between 1992 and 2012; the offenders were mainly their fathers, and they were generally murdered by highly lethal methods (firearms or fire) [26]. The explanation of this result requires subsequent in-depth studies.

Among females hospitalized due to homicide, the primary causes were fights, brawls, and rape (10,014 people (47.94%)). The classification results based on age groups showed that there were 4959 victims aged 25–44 years; they were mainly violently assaulted in marriage life or workplaces. Further analyses showed that the highest percentage of those sexually assaulted and raped were in the 5–24-year age group (65.5%), whereas female children under the age of four and elderly females over 65 years old accounted for 1.63% and 3.68% of those sexually assaulted and raped, respectively. Therefore, the targets of sexual assault by males using violent methods were mainly young girls aged 5–24 years old. Additionally, 10,014 females were assaulted or raped (E960) in this study. However, only 1.62% of patients (163/10,014) were classified as E960.1 (rape). This number clearly underestimates the actual situation. Therefore, this result shows that “sexual assault” is a very obscure issue, and many victims (females) are reluctant to expose events and use the term “violent assault.” Thus, first-line medical care personnel in the hospital emergency rooms must be vigilant to actively assist these females who are under the shadow of intimate partners to escape from sexual assault situations as soon as possible.

Furthermore, the results of this study showed that female abuse was the second most common cause of homicide hospitalization (10.86%) and was 3.60 times higher than the rate of male abuse (3.02%). Further analyses showed that female abuse occurred primarily in the 25–44-year age group (44.89%), whereas female children aged 0–4 years and elderly females aged over 65 years also accounted for 9.17% and 8.25%, respectively. One study noted that educating females in poor families makes them understand relevant legitimate rights (such as the female rights and interests act) and assisting the high-risk population to solve economic problems can prevent them from becoming homicide victims [27]. Therefore, the government should target unemployed and low-income females who are vulnerable to abuse to strengthen their ability to work and provide appropriate job opportunities that can promote their economic independence, which may reduce their chances of being abused.

### 4.3. Factors of Inpatient Mortality Due to Homicide

Our study showed that after controlling for other factors, factors associated with homicide in-hospital mortality included gender (more men than women), age (1.6% increased risk of death yearly), low income (low income was greater than non-low-income), catastrophic disease (catastrophic disease greater than no catastrophic disease), CCI score (18.4% increased risk of death per additional score), urbanization level (high risk of urbanization is greater than low urbanization), hospital level (medical center greater than regional hospitals), classification of hospitalization (risk of general medical care is greater than others), surgery (risk of surgery is greater than no surgery), cause of homicide (submersion (drowning) first, hanging and strangulation second, firearms and explosives third), and type of injury (hospitalized patients with “vascular injuries“ first, joint and muscle sprains and intracranial, chest, and abdominal pelvic injuries second, and “burns“ third, with higher risk of death). Gender patterns in injury mortality do not follow typical social justice analyses of health, in that men are at greater risk [28]. Men are more likely than women to die of almost every disease and illness and die earlier. Injury, a leading cause of premature death, was no exception [29,30]. Men’s higher unintentional injury, suicide, and homicide mortality rates are observed in all age groups in low-, middle-, and high-income countries [31]. The sole exception is the homicide of children under the age of 15 years in low- and high-income countries, where the rates for girls are similar to or higher than those for boys [28].

In 2019, an estimated 236,000 people died from drowning, making drowning a major public health problem worldwide [32]. In 2019, injuries accounted for nearly 8% of total global mortality. Drowning is the third leading cause of unintentional injury death, accounting for 7% of all injury-related deaths [32]. The true prevalence and incidence of all-cause strangulation injuries and mortality are unknown. Epidemiological studies and case series in the literature tend to be etiological and population-specific due to various etiologies that lead to common injury pathways [33]. Homicide, the most likely violent crime involving a gun, increased by nearly 27.5% nationwide in 2020 [34].

Trauma is the third leading cause of death [35]. Therefore, vascular injury plays a leading role in morbidity and mortality. Major blood vessel damage occurs in young men, usually from a stab wound. Popliteal fossa injuries, primarily caused by motor vehicle accidents, are the second most common arterial injury, followed by combined ulna and radius injuries [35,36]. In the first hours after trauma, revascularization may prevent many unnecessary and preventable amputations [35]. Pelvic ring injuries in young adults are often associated with high-energy trauma, including falls from heights and motor vehicle crashes. High-force impact means an increased incidence of related injuries to other parts of the body [37]. Burns are a global public health problem, killing an estimated 180,000 people yearly [38]. Most of these occurred in low- and middle-income countries, and almost two-thirds occurred in the WHO African and South-East Asia Regions [38]. Burn mortality has been declining in many high-income countries, and child burns mortality in low- and middle-income countries is now more than seven times higher than in high-income countries [38].

Yu et al. (2010) found that 7419 people died of homicide between 1986 and 2007, and an average of 337 people died of homicide annually [13]. Our study showed that 95,878 people were hospitalized due to homicide between 1998 and 2015 (an average of 5327 people annually), of whom 511 people died as inpatients (an average of 28 people annually). Yu’s study showed that the mortality rate among homicide events was approximately 5.95% (337/5664) [13]. Approximately 91.69% of these deaths (309/337) occurred before hospital admission or in the hospital emergency rooms [13]. Thus, the priority for homicide prevention is to strengthen the effectiveness of first aid.

### 4.4. Study Limitations

This study has several limitations. First, this study could only preliminarily analyze the causes of homicide based on HWDC and temporarily could not collect data from the population who did not seek medical treatment for analyses. Therefore, the actual risks still require further in-depth studies. Second, we could not find out if the correlates of homicide mortality in hospitals are similar to those where the victim was killed and taken to the morgue, not the hospital. Third, potential influencing factors, such as other social support and family identity should also be considered.

## 5. Conclusions

On average, 14.6 people are hospitalized due to homicide in Taiwan daily. Males and the 15–44-year age group are high-risk populations. The hospitalization rates due to homicide among all age groups have gradually decreased after the peak period in 2000. However, there has been no decreasing trend among children aged 0–4 years. Therefore, the government must strengthen prevention and control actions targeting young adults and adults. Additionally, family functions should be strengthened to reduce the incidence of child abuse or homicide.

“Fight, brawl, and rape” were the primary causes of hospitalization due to homicide. Additionally, the percentages of “fight, brawl, and rape,” “firearms and explosives,” and “cutting and piercing instruments” were significantly higher among males than females. Additionally, the percentages of “hanging and strangulation,” “corrosive or caustic substance,” “child battering and other maltreatment,” “submersion,” and “poisoning” were significantly higher among females than males. Factors associated with homicide in-hospital mortality included gender, age, low income, catastrophic disease, CCI score, urbanization level, hospital level, and causes of homicide death included drowning (drowning), hanging and strangulation, firearms, and explosives. Injury types include “vascular injury” first, joint and muscle sprain, intracranial, chest, and abdomen pelvic injuries second, and “burns” third. Therefore, government departments must strengthen interagency collaboration, especially between law enforcement/criminal justice and public health. Homicide prevention requires a comprehensive strategy beyond specific victim groups.

This issue is of great importance. Our study uses big data and is tracked long enough to be representative of Taiwanese society and what is happening in Asia and the world. Our study provides empirical data as a basis for homicide prevention and control worldwide. Nevertheless, it is simultaneously important to evaluate the latest and most accurate information. Future studies should investigate if anything has changed during the observational period (for example, comparing every ten years) from 2013 to 2022, including awareness, organizations, homicide, etc.

## Figures and Tables

**Figure 1 ijerph-19-04341-f001:**
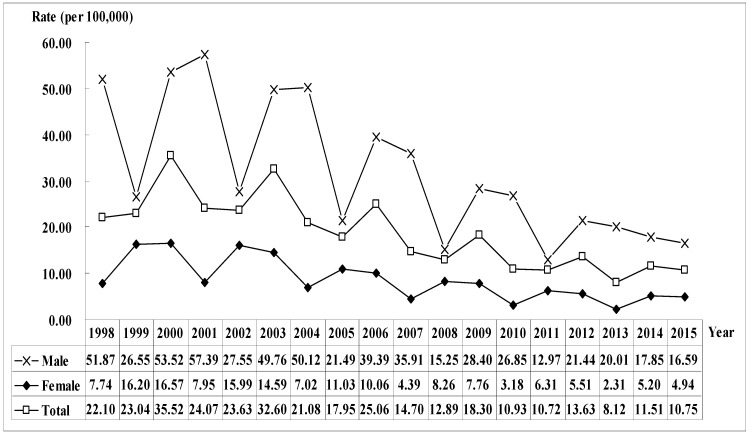
The trend of homicide hospitalization rates by gender in Taiwan, 1998–2015.

**Figure 2 ijerph-19-04341-f002:**
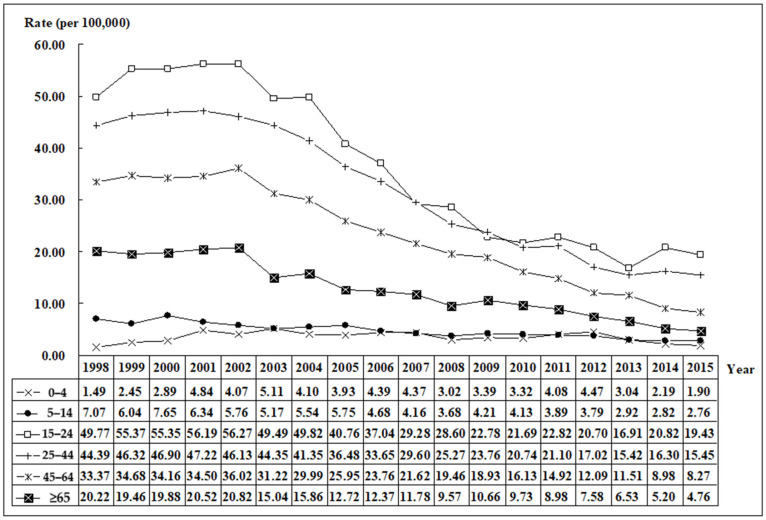
The trend of homicide hospitalization rates by age groups in Taiwan, 1998–2015.

**Table 1 ijerph-19-04341-t001:** Characteristics of homicide inpatients between 1998 and 2015 in Taiwan.

Gender	Overall		Male		Female		*p*
Variables	*n*	%	*n*	%	*n*	%	
Total	95,878		74,991	78.22	20,887	21.78	
Socioeconomic Status							
Age (years) (mean ± SD)	36.38 ± 15.67		35.28 ± 15.38		40.34 ± 16.04		<0.001
Age group							<0.001
0–4	736	0.77	426	0.77	310	1.48	
5–14	2564	2.67	2081	2.67	483	2.31	
15–24	23,141	24.14	20,492	24.14	2649	12.68	
25–44	42,916	44.76	32,958	44.76	9958	47.68	
45–64	21,466	22.39	15,605	22.39	5861	28.06	
≥65	5055	5.27	3429	5.27	1626	7.78	
Low Income							<0.001
No	93,744	97.77	73,406	97.89	20,338	97.37	
Yes	2134	2.23	1585	2.11	549	2.63	
Catastrophic Illness							0.202
Without	93,924	97.96	73,486	97.99	20,438	97.85	
With	1954	2.04	1505	2.01	449	2.15	
CCI (mean ± SD)	0.09 ± 0.43		0.08 ± 0.43		0.09 ± 0.44		0.113
Environment Factors							
Season							0.005
Spring MAR-MAY	24,117	25.15	18,717	24.96	5400	25.85	
Summer JUN-AUG	24,512	25.57	19,106	25.48	5406	25.88	
Autumn SEP-NOV	24,634	25.69	19,338	25.79	5296	25.36	
Winter DEC-FEB	22,615	23.59	17,830	23.78	4785	22.91	
Area							<0.001
North	27,481	28.66	22,392	29.86	5089	24.36	
Central	34,248	35.72	26,682	35.58	7566	36.22	
South	27,067	28.23	20,369	27.16	6698	32.07	
East	6919	7.22	5418	7.22	1501	7.19	
Outlying Islands	163	0.17	130	0.17	33	0.16	
Urbanization Level							<0.001
High	22,183	23.14	17,550	23.40	4633	22.18	
Medium	42,205	44.02	33,290	44.39	8915	42.68	
Low	31,490	32.84	24,151	32.21	7339	35.14	
Hospital Utilization							
Hospital Level							<0.001
Medical Center	16,331	17.03	13,417	17.89	2914	13.95	
Regional Hospital	43,795	45.68	34,732	46.31	9063	43.39	
District Hospital	35,752	37.29	26,842	35.79	8910	42.66	
Medical Division							<0.001
General Medical	2081	2.17	1335	1.78	746	3.57	
General Surgery	68,024	70.95	52,843	70.47	15,181	72.68	
Psychiatry	906	0.94	616	0.82	290	1.39	
Others	25,167	26.25	20,197	26.93	4970	23.79	
Surgery							<0.001
Without	60,986	63.61	45,266	60.36	15,720	75.26	
With	34,892	36.39	29,725	39.64	5167	24.74	
Hospital Admissions (mean ± SD)	1.10 ± 0.43		1.10 ± 0.43		1.09 ± 0.40		0.001
Hospitalization Days (mean ± SD)	5.86 ± 10.60		5.99 ± 11.10		5.39 ± 8.55		<0.001
Medical Expenses (USD) (mean ± SD)	983.9± 2286.4		1042.2 ± 2385.3		774.7 ± 1874.4		<0.001
Prognosis							0.017
Survival	95,367	99.47	74,569	99.44	20,798	99.57	
Mortality	511	0.53	422	0.56	89	0.43	

*p*-value (categorical variable: Chi-square/Fisher exact test; continuous variable: *t*-test) CCI = Charlson comorbidity index.

**Table 2 ijerph-19-04341-t002:** Comparison of injury types of patients hospitalized due to homicide and assault in Taiwan, 1998–2015.

Gender		Overall		Male		Female		*p*
Variables		*n*	%	*n*	%	*n*	%	
Total		95,878		74,991	78.22	20,887	21.78	
Homicide and assault causes								
E960	Fight, brawl, and rape							<0.001
	No	48,779	50.88	37,906	50.55	10,873	52.06	
	Yes	47,099	49.12	37,085	49.45	10,014	47.94	
E961	Corrosive or caustic substance							<0.001
	No	95,769	99.89	74,939	99.93	20,830	99.73	
	Yes	109	0.11	52	0.07	57	0.27	
E962	Poisoning							<0.001
	No	95,249	99.34	74,627	99.51	20,622	98.73	
	Yes	629	0.66	364	0.49	265	1.27	
E963	Hanging and strangulation							<0.001
	No	95,827	99.95	74,967	99.97	20,860	99.87	
	Yes	51	0.05	24	0.03	27	0.13	
E964	Submersion (drowning)							0.017
	No	95,852	99.97	74,976	99.98	20,876	99.95	
	Yes	26	0.03	15	0.02	11	0.05	
E965	Firearms and explosives							<0.001
	No	95,206	99.30	74,377	99.18	20,829	99.72	
	Yes	672	0.70	614	0.82	58	0.28	
E966	Cutting and piercing instruments							<0.001
	No	83,257	86.84	64,125	85.51	19,132	91.60	
	Yes	12,621	13.16	10,866	14.49	1755	8.40	
E967	Child battering and other maltreatment							<0.001
	No	91,348	95.28	72,729	96.98	18,619	89.14	
	Yes	4530	4.72	2262	3.02	2268	10.86	
E968	Others							0.316
	No	66,646	69.51	52,068	69.43	14,578	69.79	
	Yes	29,232	30.49	22,923	30.57	6309	30.21	
E969	Late Impact							0.782
	No	94,613	98.68	73,997	98.67	20,616	98.70	
	Yes	1265	1.32	994	1.33	271	1.30	
Injuries Types								
800–829	Fractures of the skull, trunk, and upper and lower limbs							<0.001
	No	65,648	68.47	49,307	65.75	16,341	78.24	
	Yes	30,230	31.53	25,684	34.25	4546	21.76	
830–839	Dislocation							0.997
	No	94,797	98.87	74,145	98.87	20,652	98.87	
	Yes	1081	1.13	846	1.13	235	1.13	
840–849	Sprains of joints and muscles							0.002
	No	94,031	98.07	73,600	98.15	20,431	97.82	
	Yes	1847	1.93	1391	1.85	456	2.18	
850–869	Intracranial, thoracic, and abdominal pelvic injuries							<0.001
	No	47,894	49.95	38,769	51.70	9125	43.69	
	Yes	47,984	50.05	36,222	48.30	11,762	56.31	
870–899	Open wounds							<0.001
	No	50,745	52.93	36,052	48.08	14,693	70.35	
	Yes	45,133	47.07	38,939	51.92	6194	29.65	
900–904	Vascular injuries							<0.001
	No	94,465	98.53	73,761	98.36	20,704	99.12	
	Yes	1413	1.47	1230	1.64	183	0.88	
910–924	Superficial skin contusions							<0.001
	No	56,359	58.78	45,743	61.00	10,616	50.83	
	Yes	39,519	41.22	29,248	39.00	10,271	49.17	
925–929	Crush injuries							0.003
	No	95,481	99.59	74,656	99.55	20,825	99.70	
	Yes	397	0.41	335	0.45	62	0.30	
930–939	Foreign body entering through natural orifice injuries							0.859
	No	95,833	99.95	74,955	99.95	20,878	99.96	
	Yes	45	0.05	36	0.05	9	0.04	
940–949	Burns							<0.001
	No	95,833	99.95	74,955	99.95	20,878	99.96	
	Yes	45	0.05	36	0.05	9	0.04	
950–957	Spinal cord injuries							<0.001
	No	92,673	96.66	72,291	96.40	20,382	97.58	
	Yes	3205	3.34	2700	3.60	505	2.42	
960–989	Drug and substance poisoning							<0.001
	No	95,073	99.16	74,598	99.48	20,475	98.03	
	Yes	805	0.84	393	0.52	412	1.97	
990–999	Others							0.510
	No	86,917	90.65	67,957	90.62	18,960	90.77	
	Yes	8961	9.35	7034	9.38	1927	9.23	

**Table 3 ijerph-19-04341-t003:** Relevant factors of inpatient mortality due to homicide and assault (n = 95,878).

Variables	Adjusted OR	95%CI		*p*
Gender				
Male	1.765	1.265	2.011	<0.001
Female	Reference			
Age	1.016	1.007	1.019	<0.001
Low Income				
Yes	3.014	2.025	4.252	<0.001
Not	Reference			
Catastrophic Illness				
With	6.741	5.023	9.021	<0.001
Without	Reference			
Charlson Comorbidity Index(CCI)	1.184	1.073	1.286	<0.001
Season				
Spring (March–May)	Reference			
Summer (June–August)	1.025	0.786	1.375	0.826
Autumn (September–November)	1.189	0.929	1.516	0.247
Winter (December–February)	1.061	0.834	1.413	0.653
Urbanization Level				
High	1.666	1.239	2.226	0.001
Medium	1.432	1.083	1.827	0.009
Low	Reference			
Hospital Level				
Medical Center	3.201	2.427	4.551	<0.001
Regional Hospital	2.603	1.918	3.452	<0.001
District Hospital	Reference			
Hospitalized Classification				
General Medical	3.986	2.604	5.822	<0.001
General Surgery	2.131	1.575	2.762	<0.001
Psychiatry	0.297	0.038	1.488	0.244
Others	Reference			
Surgery				
With	1.398	1.112	1.706	0.001
Without	Reference			
Hospitalization Days	0.996	0.984	1.001	0.052
Homicide Causes				
Fight, brawl, and rape	1.774	1.003	3.029	0.049
Corrosive or caustic substance	2.642	0.684	10.993	0.265
Poisoning	3.21	1.145	7.565	0.001
Hanging and strangulation	42.454	10.092	116.054	<0.001
Submersion (drowning)	84.901	22.901	251.787	<0.001
Firearms and explosives	5.332	2.501	10.982	<0.001
Cutting and piercing instruments	2.254	1.142	4.114	<0.001
Child battering and other maltreatment	6.513	3.685	11.982	<0.001
Others	2.001	1.086	3.598	0.008
Injuries Types				
Fractures of the skull, trunk, and upper and lower limbs	1.084	0.851	1.320	0.459
Dislocation	0.802	0.248	2.482	0.722
Sprains of joints and muscles	0.262	0.060	1.092	0.066
Intracranial, thoracic, and abdominal pelvic injuries	3.151	2.579	4.021	<0.001
Open wounds	0.422	0.318	0.533	<0.001
Vascular injuries	4.397	2.842	6.713	<0.001
Superficial skin contusions	0.211	0.158	0.300	<0.001
Crush injuries	0.534	0.062	3.989	0.572
Foreign body entering through natural orifice injuries	No events	-	-	0.999
Burns	2.896	1.423	5.131	<0.001
Spinal cord injuries	0.334	0.108	0.704	0.001
Drug and substance poisoning	1.372	0.655	2.836	0.382
Others	2.121	1.642	2.711	<0.001

## Data Availability

The data that support the findings of this study are available from the Health and Welfare Data Science Center, Ministry of Health and Welfare (HWDC, MOHW) but restrictions apply to the availability of these data, which were used under license for the current study, and so are not publicly available. Data are, however, available from the authors upon reasonable request and with permission of HWDC (http://www.mohw.gov.tw/cht/DOS/DM1.aspx?f_list_no=812 (accessed on 15 January 2022)).

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
