# Peer review of "Trends in Homicide Hospitalization and Mortality in Taiwan, 1998–2015"

_ijerph, 2022, doi:10.3390/ijerph19074341_

Round 1

Reviewer 1 Report

Certainly, the topic of the article is important and interesting. The survey is based on basic statistical methods, so it is not methodologically advanced. As the methods used are basic, the asset of the article should be rich and interesting conclusions. In part 4 (Discussion) it is indeed so, but part 3 (Results) requires a lot of elaboration in terms of descriptions and conclusions.

Detailed remarks:

In the description of the literature (lines 50-67), it is necessary to indicate who and when published this research, then this part of the introduction will be more understandable.

At the end of the introduction, it is worth mentioning what elements the research consists of and what they contain (organization of the article).

Page 4, Table 1 - it would also be worth pointing to the dominant feature in the statistical analyses.

Table 1 would be more readable if Overall, Male and Female were separated by a line.

Table 1 contains a lot of information (a lot of data) that have not been discussed at all in the text - this should be corrected, the reader cannot analyze this table himself, conclusions should be provided by the authors of the research.

Table 2 - the description of the data contained in the table is also too poor, it should be extended. Pay attention to the most interesting results, describe them accurately - maybe extend the descriptions and look for reasons why this is so? Again, the reader cannot analyze the data on his own, he must be given deep, interesting conclusions.

Table 3 - the authors should think more deeply about the description, on the basis of such a large amount of data, many more conclusions can certainly be drawn.

Part 4 of the article (Discussion) is very interesting. Part 3 (Results) should be written similarly, now it lacks deeper analysis in conclusions. Due to the fact that the work is not methodologically advanced, its advantage must be the descriptions and conclusions drawn from the data.

In my opinion, the summary is too short. It should be extended, and it is also worth giving directions for further research.

Reviewer 2 Report

The following comments/concerns are not in any order.

  1. There is a censoring problem with the data because homicides where the victim was already dead are not included in the analyses since the focus is on hospital mortality. We have no way of knowing if the correlates of homicide mortality in hospital are similar to those where the victim was killed and taken to the morgue, not hospital. This has to be addressed substantively as a limitation and if comparative data are available, would be a good supplemental analysis.
  2. The literature review is scant and omits coverage of other studies that show how medical access affect homicide mortality. The following works can help toward this end:
  3. Doerner, W. G., & Speir, J. C. (1986). Stitch and sew: The impact of medical resources upon criminally induced lethality. Criminology24(2), 319-330.
  4. Doerner, W. G. (1988). The impact of medical resources on criminally induced lethality: A further examination. Criminology26(1), 171-180.
  5. Harris, A. R., Thomas, S. H., Fisher, G. A., & Hirsch, D. J. (2002). Murder and medicine: the lethality of criminal assault 1960-1999. Homicide studies6(2), 128-166.
  6. Summers, L., & Rogers, T. G. (2020). Too far for comfort? Situational access to emergency medical care and violent assault lethality. Crime Science9(1), 1-11.
  7. Hatten, D. N., & Wolff, K. T. (2020). Rushing gunshot victims to trauma care: The influence of first responders and the challenge of the geography. Homicide studies24(4), 377-397.
  8. The findings are largely consistent with other research on demographic correlates of homicide. Can the author identify unique ways these findings are important in a Taiwan context.

Reviewer 3 Report

This study may give implications for domestic clinical research; however, what could be implications for international audience? In addition, what this paper reveals cannot be translated into a message that the authors claimed in the abstract. The paper should have a sharp focus and clear research questions. Especially, why is this kind of study important and to whom?

Round 2

Reviewer 1 Report

The authors have improved the article taking into account the comments from the review. I think that the descriptions of the results presented in the tables could still be worked on, nevertheless it looks much better now. 

The authors seemed to be in a hurry correcting the article, because there are some editing mistakes in the text, they should read the text carefully and correct them.

Reviewer 2 Report

The authors adequately responded to my concerns.
